# Multimodal Diagnostics of Changes in Rat Lungs after Vaping

**DOI:** 10.3390/diagnostics13213340

**Published:** 2023-10-30

**Authors:** Irina Yu. Yanina, Vadim D. Genin, Elina A. Genina, Dmitry A. Mudrak, Nikita A. Navolokin, Alla B. Bucharskaya, Yury V. Kistenev, Valery V. Tuchin

**Affiliations:** 1Institution of Physics, Saratov State University, 410012 Saratov, Russia; versetty2005@yandex.ru (V.D.G.); eagenina@yandex.ru (E.A.G.); tuchinvv@mail.ru (V.V.T.); 2Laboratory of Laser Molecular Imaging and Machine Learning, Tomsk State University, 634050 Tomsk, Russia; allaalla_72@mail.ru (A.B.B.); yuk@iao.ru (Y.V.K.); 3Science Medical Center, Saratov State University, 410012 Saratov, Russia; 4Department of Pathological Anatomy, Saratov State Medical University, 410012 Saratov, Russia; xupypr-wh@mail.ru (D.A.M.); nik-navolokin@yandex.ru (N.A.N.); 5Experimental Department, Center for Collective Use of Experimental Oncology, Saratov State Medical University, 410012 Saratov, Russia; 6State Healthcare Institution, Saratov City Clinical Hospital No. 1 Named after Yu.Ya. Gordeev, 410017 Saratov, Russia; 7Institute of Precision Mechanics and Control, FRC “Saratov Scientific Centre of the Russian Academy of Sciences”, 410028 Saratov, Russia

**Keywords:** lung, vape, optical properties, OCT, histology

## Abstract

(1) Background: The use of electronic cigarettes has become widespread in recent years. The use of e-cigarettes leads to milder pathological conditions compared to traditional cigarette smoking. Nevertheless, e-liquid vaping can cause morphological changes in lung tissue, which affects and impairs gas exchange. This work studied the changes in morphological and optical properties of lung tissue under the action of an e-liquid aerosol. To do this, we implemented the “passive smoking” model and created the specified concentration of aerosol of the glycerol/propylene glycol mixture in the chamber with the animal. (2) Methods: In ex vivo studies, the lungs of Wistar rats are placed in the e-liquid for 1 h. For in vivo studies, Wistar rats were exposed to the e-liquid vapor in an aerosol administration chamber. After that, lung tissue samples were examined ex vivo using optical coherence tomography (OCT) and spectrometry with an integrating sphere. Absorption and reduced scattering coefficients were estimated for the control and experimental groups. Histological sections were made according to the standard protocol, followed by hematoxylin and eosin staining. (3) Results: Exposure to e-liquid in ex vivo and aerosol in in vivo studies was found to result in the optical clearing of lung tissue. Histological examination of the lung samples showed areas of emphysematous expansion of the alveoli, thickening of the alveolar septa, and the phenomenon of plasma permeation, which is less pronounced in in vivo studies than for the exposure of e-liquid ex vivo. E-liquid aerosol application allows for an increased resolution and improved imaging of lung tissues using OCT. Spectral studies showed significant differences between the control group and the ex vivo group in the spectral range of water absorption. It can be associated with dehydration of lung tissue owing to the hyperosmotic properties of glycerol and propylene glycol, which are the main components of e-liquids. (4) Conclusions: A decrease in the volume of air in lung tissue and higher packing of its structure under e-liquid vaping causes a better contrast of OCT images compared to intact lung tissue.

## 1. Introduction

Respiratory diseases are a public health problem around the world. According to the World Health Organization, 235 million people suffer from asthma, and more than 200 million people have chronic obstructive pulmonary disease [1], which determines the relevance of developing new methods for the diagnostics of lung diseases [2,3,4].

An endoscopic lung imaging technique combines the advantages of whole-lung structure imaging with the ability of a morphological examination to reveal a fine subcellular structure. Modern non-invasive imaging methods such as ultrasonography, computed tomography (CT), magnetic resonance imaging (MRI), and positron emission tomography (PET) allow detailed and dynamic 3D analysis of anatomical structures in living organisms, but these methods have a limited spatial resolution, which is often not sufficient to visualize tissue and cell microstructures [5,6,7,8,9,10,11,12,13,14,15,16,17,18,19]. Optical coherence tomography (OCT) has considerable potential for lung microstructure analysis [20,21,22]. However, the main limitation of OCT applications for lung tissue is a low light penetration depth. To increase the contrast of OCT images, Kohlfaerber et al. used a microscopic resolution OCT (mOCT) with temporal tissue fluctuation (dynamic mOCT) for visualizing the morphological and functional micro-anatomy of the airways in vivo [23]. Schnabel et al. compared three tissue optical clearing (TOC) methods to track the lung bio-distribution of hUC-MSCs labeled with the tdTomato fluorescent protein [24]. The direct detection of the tdTomato cells was only possible using the CUBIC clearing protocol, which resulted in highly transparent lungs and allowed for studying the interaction of the hUC-MSCs with cells in the host’s lung. The use of machine learning in data processing has gained great development [3,25,26,27,28]. A promising approach to increase the depth and contrast of OCT tissue imaging is to apply a TOC technique [29,30,31,32].

TOC, in combination with molecular labeling and optical sectioning microscopy, has become an important tool for 3D imaging in biological applications, including the investigation of cell bio-distribution in whole organs [33]. Despite the many advantages, practical limitations to the applicability of TOC protocols for imaging still remain under analysis. Several protocols have been developed [34], and a choice of a proper one requires the careful consideration of a range of parameters to achieve the optimal trade-off for every case. Sample size and tissue composition impact the TOC speed and limit the suitable microscopy tools [35]. Fluorophore preservation poses another challenge, as certain TOC protocols are incompatible with fluorescent probes [36]. In particular, the preservation of protein-based fluorescence and lipid staining remains a challenge [37]. The compatibility of immunostaining with the chemicals used for TOC, as well as antibody penetration in large samples, requires testing and optimization [35]. Moreover, certain parameters depend on tissue and sample types, and no single TOC approach fits all, which requires the use of application/tissue-specific protocols.

During the last few decades, optical clearing agents have been deeply explored and largely exploited for tissue TOC, allowing deeper light penetration into optically inhomogeneous media [38]. Obviously, one of the major goals of all the research activities carried out up to now in this field consists of translating the TOC method at the clinical level and in vivo TOC of biological tissues. When moving tissue TOC from ex vivo to in vivo, several issues have to be taken into consideration. In particular, safety and biocompatibility requirements drastically limit the panorama of TOC agents, which can be employed in vivo in humans. Basically, the set of TOC agents suitable for this purpose is limited. In fact, the need for a reversible TOC process confines the choice to aqueous solutions of glycerol, sugars, polyethylene glycol, propylene glycol, or acetic acid, excluding any other compound because of toxicity and/or chemical aggressiveness. Thus, as an optical clearing agent, a mixture of glycerol and propylene glycol used in e-cigarette liquids can be helpful.

In this work, the inhalation of an e-liquid was studied as a potential diagnostic test to examine pathological changes in lung tissue in vivo and ex vivo. To do this, we implemented a “passive smoking” model and created the specified concentration of aerosol of a glycerol/propylene glycol mixture in the chamber with the animal.

## 2. Materials and Methods

### 2.1. E-Cigarette Liquid and Experimental Setup

We implemented the model of “passive smoking”, in this model, the patient receives up to 20% of the concentration of toxic substances of an active smoker. Thus, our goal was to create a similar concentration of glycerol mixture vapor in the chamber with the animal.

Aerosol of e-cigarette nicotine-free e-liquid consisting of glycerol (60%) and propylene glycol (40%) (Alliance, Moscow, Russia) was used. It is known that these components have low toxicity and are authorized for use in medicine and cosmetic industry [39,40]. Refractive index of the liquid was measured with a multi-wavelength Abbe DR-M2/1550 refractometer (Atago, Tokyo, Japan) in the wavelength range from 480 to 1550 nm at room temperature 23 °C.

The experimental setup consisted of an inhalation chamber (Figure 1) with a special device for aerosol pumping. A 150 mL Janet syringe is attached to the Vaperi Just 3 (Shenzhen Eleaf Electronics Co., Shenzhen, China). When the vaper generates vapor, it is drawn into the syringe and injected into the inhalation chamber.

The design of the inhalation chamber involved diluting the e-cigarette aerosol with air. Thereby, vape inhalation was simulated when the predicted deposition of e-cigarette aerosol in the lungs was not more than 25% [41]. To develop the intensive vaping test for lung optical clearing, 1 h exposure to aerosol of nicotine-free e-liquid was used.

### 2.2. Animal Study Protocols and Lung Samples Preparation

The in vivo study was performed on 12 sexually mature male Wistar rats weighing 200 ± 20 g. The animals were divided into two groups—experimental and control—with 6 rats in each. The rats in the control group have not been subjected to vaping. The rats of the experimental group were placed in the inhalation chamber, and e-cigarette liquid aerosol was pumped into it at the rate of 0.15 L/min.

About 3 mL of e-liquid was used in the study. There were 5 exposures of animals being placed in the inhalation chamber. On average, 0.6 mL of e-liquid was consumed per exposure. The volume of aerosol introduced into the inhalation chamber was 1500 mL; therefore, the concentration of e-liquid in the aerosol was 0.004%. In the chamber, aerosol was diluted with air to 7500 mL (5 times, 20% concentration of primary vapor in the air inhaled by the animal); therefore, the concentration of e-liquid in the chamber was 0.0008%. Aerosol with such a concentration of glycerol/propylene glycol mixture was inhaled by a rat. The average minute volume of the lungs of a Wistar rat weighing about 363 g is 303 mL/min [42]. The rat spent 10 min in the chamber, which means that during each exposure, it inhaled 3030 mL of air, and during all 5 exposures, it inhaled 15,150 mL. Thus, the volume of e-liquid that passed through the rat’s lungs during the entire experiment is 0.12 mL.

According to the model of deposition of e-liquid in a person’s lungs [41], during passive smoking through the nose, 7–10% of the aerosol settles in the lungs. The total respiratory surface of the lungs of a white laboratory rat with weight of 363 g is equal to 7.5 m^2^ [43]. Thus, the thickness of the layer of e-liquid deposited on the surface of the rat’s lungs is 1.1–1.2 µm.

The animals were removed from the experiment one hour after inhalation, and their lungs were taken. Twenty samples were collected from each group for the next step of study.

In ex vivo studies, lungs of 6 Wistar rats were placed in the e-liquid for 1 h. After that, twenty samples were cut and examined using the imaging technique.

### 2.3. The Spectroscopy and Imaging Techniques

The spectra of total transmittance and diffuse reflectance of the samples were measured in the wavelength range of 350–2500 nm using a UV-3600 spectrophotometer with an integrating sphere LISR-3100 (Shimadzu, Kyoto, Japan). Before measurements, the samples were placed between two glass slides and fixed without compression, which guaranteed absence of changes in tissue morphology. The thickness of the samples was measured by a micrometer with an accuracy of ±1 µm at five points of the sample, after which the results were averaged and the standard deviation was calculated. Average thickness of the samples was 0.50 ± 0.13 mm. The absorption μ_a_ and reduced scattering $\mu_{s}^{\prime }$ coefficients of the lung tissue were calculated using the inverse adding-doubling method [44,45].

OCT B-scans were recorded using the Spectral Radar OCT system (Thorlabs, Newton, NJ, USA) with a central radiation wavelength of 930 nm, a spectral bandwidth of 100 nm, a depth resolution in air of 6.2 μm, a lateral resolution of 8.3 μm, an optical scanning depth in the air of 1.6 mm, and scanning length along tissue surface of 6 mm.

3D OCT images were measured using the OCT GAN930V2-BU (Thorlabs, Newton, NJ, USA) with a center wavelength of 930 nm, a spectral bandwidth of 150 nm, longitudinal and lateral resolutions of 6.0 and 7.3 μm, respectively. Based on the recorded OCT tomograms with entire scanning length along tissue surface of 6 mm, the coefficient of light attenuation in the tissue was calculated by approximating the dependence of the reflected light intensity *I(z)* on the depth of the studied area *z* of the A-scan: $I\left(z\right)=A_{D}\exp\left\{-\mu_{t}z\right\}+y_{0}$. The attenuation coefficient was evaluated using the Mathcad 14 software.

### 2.4. Histopathological Analysis

The lung tissue samples of thickness from 1.5 to 2.5 mm were taken by surgery from rats. For histological examination of excised tissue samples, fixation was carried out by a 10% solution of formaldehyde. After fixation, the 5 to 7 μm tissue slices of lung tissue were made, stained by hematoxylin-eosin (H&E) using a standard technique, and then analyzed [46]. Morphological study was conducted independently by two experts in the field of pathological anatomy: a well-experienced postgraduate student and a professor, doctor of medical sciences. Thus, the study was suited to a double-blind manner.

## 3. Results and Discussion

Histological examination of samples conducted by in vivo exposure to e-cigarette liquid showed areas of emphysematous expansion of the alveoli (Figure 2b), thickening of the alveolar septa, which is partly due to erythrocyte diapedesis (Figure 2c), and the phenomenon of immersion liquid permeation (Figure 2d), which is less pronounced than the exposure ex vivo (see Figure 3).

Histological examination of ex vivo samples revealed the following morphological changes in the lung tissue structure. In the lungs of animals, areas of atelectasis predominated (Figure 3a), and a pronounced phenomenon of immersion liquid permeation (Figure 3b) was observed. The thickening of the inter-alveolar septa was probably dueto infiltration with cellular elements and e-cigarette liquid permeation (Figure 3c). There are signs of desquamation of the bronchial wall epithelium (Figure 3d).

According to morphometry in thein vivo and ex vivo studies, the average thickness of the inter-alveolar septa was 0.05 ± 0.02 mm and 0.03 ± 0.01 mm, respectively. The average thickness of the inter-alveolar septa in the control group was 0.015 ± 0.05 mm, which is consistent with the literature data [47,48]. The total lung capacity of the rat was about 10 mL compared to about 1 mL of the mouse and 6000 mL of a human [49].

In ex vivo experiments, the biological tissue was directly soaked in nicotine-free liquid. Steam collected in droplets and, for some time, was on the inner surface of the alveoli and acted on the tissuein in vivo experiments. In this sense, the action may be similar in ex vivo and in vivo experiments.

Atelectasis observed under ex vivo conditions is caused by the fact that physiological respiration is absent in the lungs removed from the organism, which results in the processes of lung tissue decay and compaction. Histological images of control lung samples in in vitro experiments appear similar to those in in vivo experiments (Figure 2a).

The health effects of exposure to e-cigarettes, especially chronic exposure, are uncertain. In most cases, e-liquid toxicity is due to the flavorings and nicotine in it. Actually, e-cigarettes emit volatile carbonyls, reactive oxygen species, furans, and metals (nickel, lead, chromium), many of which are toxic to the lungs [50,51].

Propylene glycol and vegetable glycerol are “generally recognized as safe” if added in recommended amounts to food. However, this label does not apply to inhalational safety, and occupational exposures to propylene glycol caused irritation and either mild or no objective effects on pulmonary function, suggesting that it may act as a sensory irritant [52,53,54]. The increased mucin expression was found in the primary airway epithelia in e-cigarette users [55]. It was accompanied by decreased membrane fluidity in the airway epithelia, which may affect endocytosis, exocytosis, and plasma membrane protein–protein interactions. Thus, to further develop the intense vape test for optical clearing of lung tissue, a determination of the minimum effective and safe doses of inhaled e-liquid vapor is necessary.

A set of 3D OCT images of lung tissue for intact animals after ex vivo and in vivo exposure to e-liquid is shown in Figure 4. A decrease in the volume of air in the lung tissue and the packing of its structure can cause a change in OCT signals, in contrast to intact lung tissues (see Figure 5). Also, in both cases, we observed a thickening of the inter-alveolar septa, owing to the phenomenon of plasma permeation in ex vivo conditions and infiltration by cellular elements in in vivo conditions.

In an intact lung, air fills the alveoli. They are clearly visible in OCT images in the form of dark (none scattering) objects (Figure 4a and Figure 5a) and are also observed in histological sections of lung tissue as white objects. Owing to atelectasis of the lung tissue, the air leaves the alveoli, and they are filled with agent fluid. Thus, the tissue became more transparent, which is clearly visible in the ex vivo group (see Figure 4b and Figure 5b). In the in vivo group, the expansion of the alveoli is observed, and the effect of the optical clearing is less pronounced (Figure 4c and Figure 5c).

The calculated attenuation coefficient values based on the results of OCT B-scanning at a wavelength of 930 nm are μ_t_ = 49.6 cm^−1^ (control rat lung tissue (Figure 5a)); μ_t_ = 14.0 cm^−1^ (after ex vivo exposure to e-liquid (Figure 5b)); μ_t_ = 24.6 cm^−1^ (after in vivo exposure to e-liquid (Figure 5c)). Because of the lack of data on the optical properties of rat lungs in the literature, we have compared these results with data measured for intact human lung tissue at 1064 nm: μ_a_ = 2.8 cm^−1^; μ_s_ = 39 cm^−1^ [56]. Thus, μ_t_ = μ_a_ + μ_s_ = 41.8 cm^−1^, which is in good agreement with our result.

Thus, ex vivo and in vivoexposure by the e-liquid consistingof glycerol and propylene glycol was found to result in highly effective lung TOC with a 2–3-fold decrease in the attenuation coefficient. Moreover, the changes in lung tissue structure after the application of e-liquid during 50 min of vaping are well seen in the OCT images.

The biological meaning of the proposed test is to provide the lung TOC to enhance facilities of the follow-up OCT of ex vivo lungs and to prove that a short time vaping may help in the future to improve significantly endoscopic optical modalities for lung studying, including endoscopic OCT [24,32]. TOC allows for increased resolution and improved imaging of lung tissues using OCT and many other optical techniques [57].

Initially, we tested the effects of vape liquid on the lungs of animals ex vivo, and the clearing effect was found after placing the lungs in e-liquid for 1 h. Then, we conducted in vivo experiments to study the efficacy and safety of a 1 h intense vape exposure in animals. These in vitro and in vivo experiments are not relevant to people who really enjoy vaping but are aimed at developing a diagnostic protocol to improve the visualization of lung tissue by OCT.

The spectral dependences of total transmittance and diffuse reflectance averaged over the studied groups of animals are shown in Figure 6. The shape of the spectra is determined by the absorption of the main tissue chromophores (hemoglobin, lipids, and water) and the strong scattering by the components of the lung parenchyma (alveoli, alveolar ducts, and bronchioles) filled by air and other scatterers, such as cellular elements [32,58] is observed. A large standard deviation is caused by a deviation in the thickness of the samples.

The reconstructed spectra for absorption and reduced scattering coefficients using data presented in Figure 6 are shown in Figure 7. The absorption bands of deoxygenated hemoglobin at wavelengths of 428 and 555 nm, water bands at wavelengths of 1452 and 1924 nm, and lipids band at 1748 nm are marked on the absorption spectra (Figure 6a). Single symbols show data from Refs. [56,59,60], where frozen samples of human lungs [56] and fresh samples of rabbit and piglet [59] were measured at several wavelengths using integrating spheres. Ex vivo porcine samples were measured using a broadband time-domain diffuse optical spectroscopy [60]. It is clearly seen that the results of our measurements of the lung absorption coefficient and the literature data are in good agreement. We can also see significant differences between the control group and the ex vivo group in the spectral range of water absorption. It can be associated with dehydration of lung tissue owing to the hyperosmotic properties of glycerol and propylene glycol [61].

The large variation in the scattering of samples within groups in our experiments and the values of $\mu_{s}^{\prime }$ presented in the literature data can be caused by different states of the measured tissues (Figure 7b). For example, in Ref. [62], it was shown that the optical properties of lung tissue are determined as a function of lung volume. At the measurements in vitro, the samples collapsed, and the alveoli were filled with saline instead of air. Therefore, the scattering decreased. It is an uncontrolled process, which depends on the sample’s storage conditions before measurements, preparation procedures, and measurement methods.

In Figure 7b, significant changes are observed for the reduced scattering coefficient spectra of the lung tissues after exposure to the e-liquid (ex vivo group). One can clearly see a decrease in the $\mu_{s}^{\prime }$ values in comparison with the control values associated with the TOC effect. In the ex vivo group, the samples were placed in an immersion solution with avolume larger thanthe volume of the tissue sample. In this case, two oppositely directed fluxes of water out and immersion agents into the tissue took place. Replacement of interstitial water with an immersion agent led to the occurrence of tissue dehydration and a matching tissue components refraction index (RI).

The mean RI value of the lung in the spectral range of 456–1550 nm was 1.38–1.35 [57,63]. Diffusion of the TOC agent with RI = 1.46 (589 nm) in the interstitial space and the filling of the alveoli increases the mean RI, bringing it closer to the RI of the connective tissue (alveolar septa) and epithelium [64]. The water loss induces a decrease in the thickness and an increase in the packing density of the tissue scatterers inside the sample [31]. Phenomena such as atelectasis and plasma impregnation of lung tissue, detected on histological samples in this group, also contributed to the reduction in optical inhomogeneities of the lung tissue. A decrease in the slope of the reduced scattering spectra at TOC can be associated with a decrease in the concentration of scatterers of aparticular (small) size in the tissue due to the full immersion effect.

In the in vivo group, there was an increase in scattering, which can be caused by the expansion of the alveoli and thickening of the alveolar septa. At the same time, the immersion effect was insignificant, owing to the short period of action and the absence of the RI matching effect.

For the OCT data, it was found that the initial attenuation coefficient was μ_t_ = 49 cm^−1^, and after the in vivo action of e-liquid, it was approx. twice less at μ_t_ = 24 cm^−1^. When comparing these data with spectral measurements, we can suppose that the g-factor was decreased during in vivo action because the reduced scattering coefficient was increased. Namely, effective Mie scatterers, i.e., large particles providing high anisotropy factors, turn into Rayleigh scatterers during in vivoimmersion in the e-liquid since the alveoli are not wholly filled with the agent, and some inflammation of the lung epithelial layer may occur [65,66]. This effect can also be related to the optical clearing of the inter-alveolar connective tissue, which is seen in OCT only as it is sensitive to the improvement of the ballistic photon flux caused by a reduction inthe scattering coefficient associated with the appearance of smaller effective scatterers, as diffuse transmittance and reflectance of tissue depend on a reduced scattering coefficient, which is increased slightly due to the reduction in the anisotropy factor, which is also caused by the appearance of less-effective scatterers.

According to OCT, there is always an under estimation of the scattering coefficient, which is apparently due to the fact that not only ballistic but also quasi-ballistic photons are involved in image formation.

Thus, lung TOC is recorded reliably in OCT but is poorly recorded regarding total transmittance and diffuse reflectance measurements. Since the structure of the lung tissue is highly heterogeneous, the local TOC visible during OCT does not manifest itself during the measurements in the integrating sphere, which measures the signal from the bulk tissue.

## 4. Conclusions

Multimodal diagnostics of changes in optical properties of rat lungs after vaping wereconducted. OCT and spectral measurements, together with histological analysis of lung samples, were carried out. Absorption and reduced scattering coefficients of lung samples were calculated for control and experimental groups.

Ex vivo and in vivo exposure to e-cigarette nicotine-free liquid consistingof glycerol and propylene glycol was found to result in lung TOC. Histological examination of samples showed areas of emphysematous expansion of the alveoli, thickening of the alveolar septa, and the phenomenon of plasma permeation, which is less pronounced in vivo than with ex vivoexposure. A decrease in the volume of air in the lung tissue and an increase in the packing density of its structure under vaping led to better contrast of OCT images compared to intact lung tissue.

The developed TOC protocols can be used in the future for in vivo lung studies using endoscopic OCT and other endoscopic optical imaging techniques. The minimal harmfulness of such protocols in in vivo studies follows from the relatively short-term exposure to aerosol from well-purified commercial e-liquids.

## Figures and Tables

**Figure 1 diagnostics-13-03340-f001:**
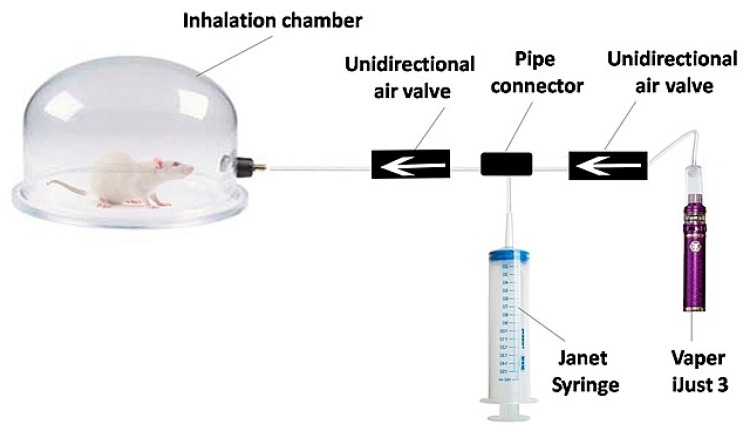
Scheme of the experimental setup for in vivo inhalation exposure.

**Figure 2 diagnostics-13-03340-f002:**
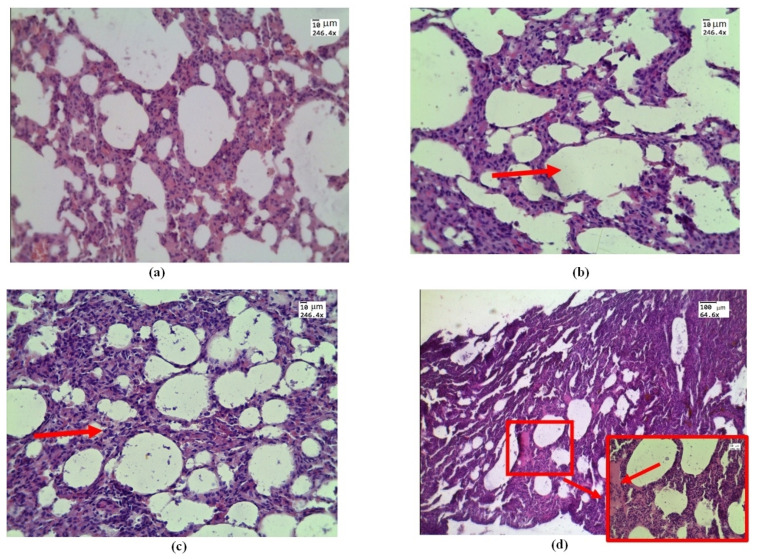
Morphological changes in rat lung tissue under in vivo exposure of e-liquid of 7.5 L of aerosol during 50 min: (**a**) control—lung without treatment; (**b**) area of emphysematous expansion of the alveoli after aerosol exposure (red arrow); (**c**) thickening of the alveolar septa (red arrow); (**d**) the phenomenon of plasma (red arrow and box). H&E. Magnification: (**a**–**d**)—246.4; and (**d**)—64.6.

**Figure 3 diagnostics-13-03340-f003:**
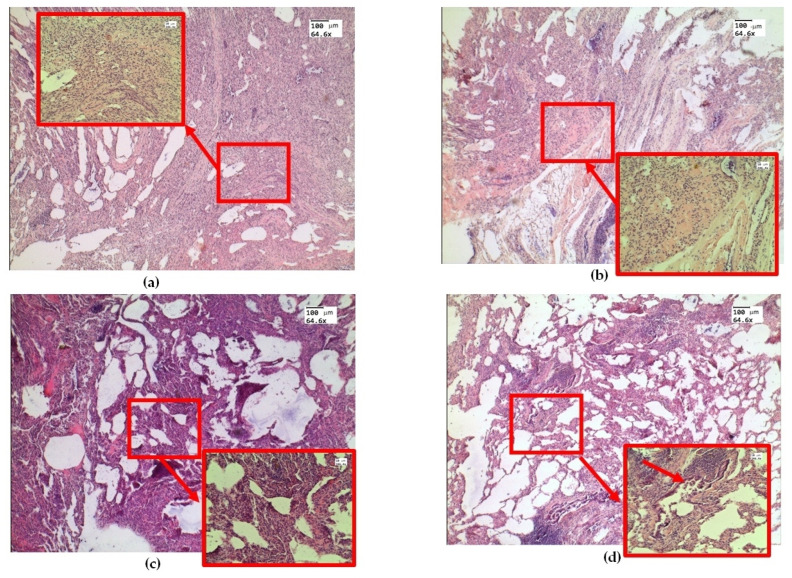
Morphological changes in ex vivolung tissue after exposure to e-liquid: (**a**) atelectasis (red arrow and box); (**b**) immersion fluid permeation in lung tissue (red arrow and box); (**c**) thickening of the inter-alveolar septa (red arrow and box); (**d**) signs of desquamation of the bronchial epithelium (red arrow and box). H&E. Magnification: 64.6 and 246.4.

**Figure 4 diagnostics-13-03340-f004:**
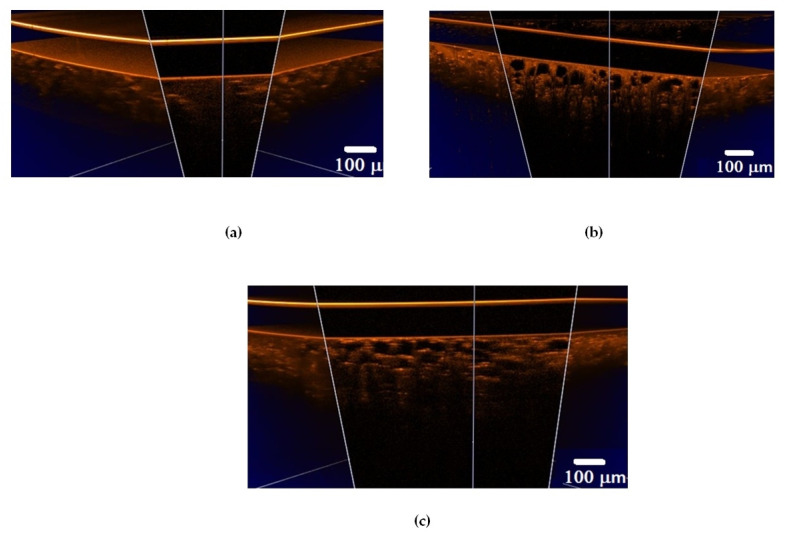
Three-dimensional OCT B-scans of ex vivo lung tissue samples: (**a**) intact animal; (**b**) after exposure to e-liquid of ex vivo sample; (**c**) after in vivo exposure to e-liquid. The sample was placed between a glass slide and a cover glass. The average geometric thickness of the cover glass was 0.15 mm and the refractive index was 1.5.

**Figure 5 diagnostics-13-03340-f005:**
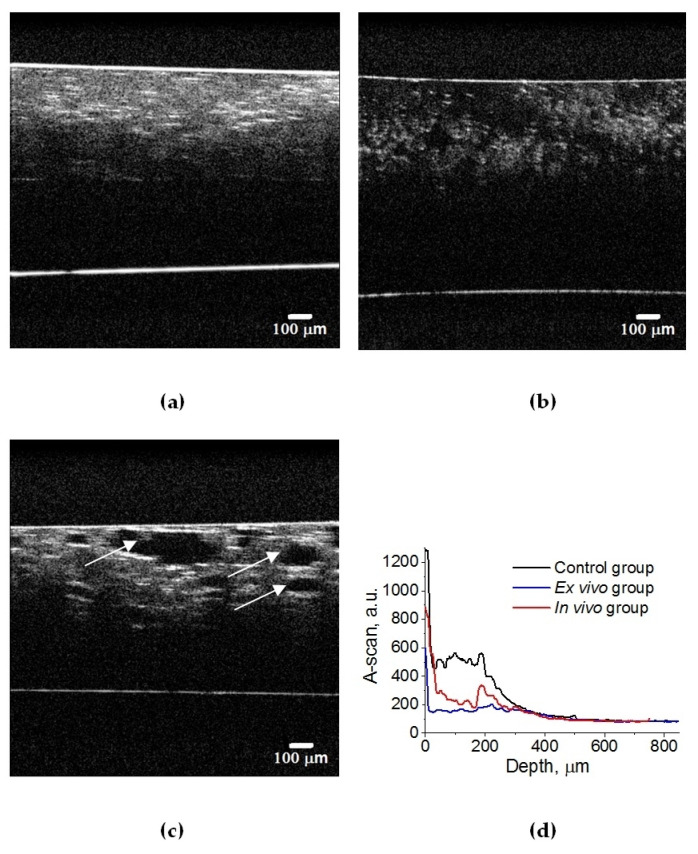
OCT B-scans of lung tissue: (**a**) intact animal; (**b**) after ex vivo exposure to e-liquid; (**c**) after in vivo exposure to e-liquid; and (**d**) corresponding average A-scans. The white arrows indicate lung alveoli filled with immersion e-liquid, resulting in a reduction in light scattering.

**Figure 6 diagnostics-13-03340-f006:**
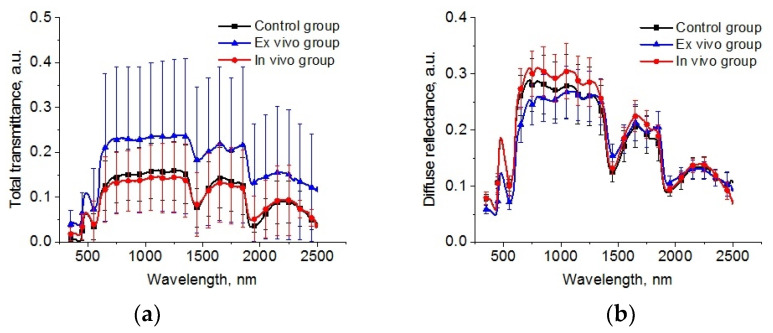
The spectra of total transmittance (**a**) and diffuse reflectance (**b**) of the intact lung samples (control group), samples after 1 h exposure in the e-liquid (ex vivo group) and samples from animals with inhalation of the e-liquid aerosol during 50 min (in vivo group).

**Figure 7 diagnostics-13-03340-f007:**
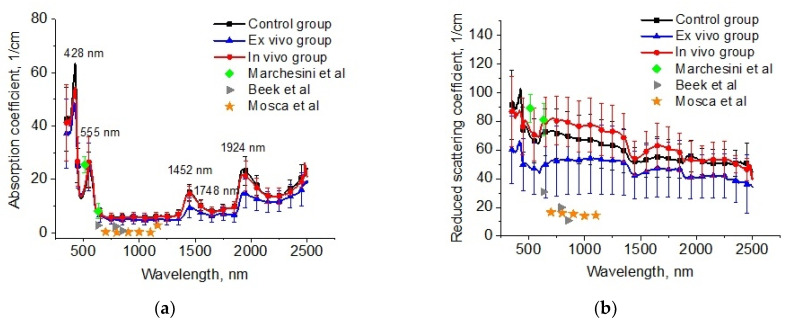
The spectra of absorption (**a**) and reduced scattering (**b**) coefficients of the intact lung samples (control group), samples after 1 h exposure in the e-liquid (ex vivo group) and samples from animals with inhalation of the e-liquid aerosol during 50 min (in vivo group). Single symbols correspond to the optical properties of in vitro lung tissue from Refs. [56,59,60].

## Data Availability

The data presented in this study are available on request from the corresponding author. The data are not publicly available due to privacy or ethical restrictions.

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
