# Peer review of "Multimodal Diagnostics of Changes in Rat Lungs after Vaping"

_diagnostics, 2023, doi:10.3390/diagnostics13213340_

Round 1

Reviewer 1 Report

Comments and Suggestions for Authors

This is an interested study using different model to study the lung morphology after vaping, which is also performing in both in vivo and ex vivo. The introduction and discussion have organized accordingly for the focused topic. However, please explained that what is the biological meaning of doing 1 day extensive vaping and then perform the analysis. also, Ex vivo model said that: " lungs were placed in the e-cigarette liquid for 1 hour" which has less biological/translational meaning to the people who really vaping. Also, based on the exposure system, is there any air dilution in between? or the lungs are majorly damaged because of CO toxicated or other metals/chemicals released from the e-cig products? 

Minor comment:

1.FIgure 7 panels are overlapping, please fix it.

2. Method sections should be including the imaging steps, and staining steps, not everything included in animal study and lung preparation section

Author Response

Dear Editor,

We thank referee for thorough review of our manuscript. We are very thankful for the effort invested by the referee, for the constructive suggestions, and we feel that the quality of our paper has been substantially improved with the corresponding changes. We agree with comments of the reviewer. We did changes suggested by the reviewer. All changes are highlighted in the paper text.

Best regards, Irina Yanina

Reviewer 2 Report

Comments and Suggestions for Authors

In this manuscript, the authors used histopathological analysis, OCT, and spectral measurement to evaluate changes in the rat lung after vaping; here are some suggestions for the author that might improve the quality of their manuscript:
1- The abstract should be restructured, avoid extensive description of the methodology.
2- The experimental methods aren’t clear; I wonder if the control rats were treated with air or left without treatment?; Also, exposing rats to 7.5 liters in 50 minutes is excessive for their lungs.
3- From the histopathological pictures, it seems that lung tissue processing wasn’t properly performed (lungs needs to be gently inflated with fixative after harvesting them from animals); as it’s hard to interpret the histopathological findings. In figure 2d and all figure 3, it will be great to show lower magnification and higher magnification picture, so the reader will have idea in which part of the lung has the alterations. In figure 3d, I could not see any bronchial epithelium, can authors provide a clearer picture?

Comments on the Quality of English Language

NA

Author Response

(The authors gave the same response as above.)

Round 2

Reviewer 2 Report

Comments and Suggestions for Authors

The authars improved their manuscript quality

Author Response

Response to Editor Comments

Point 1: In the results subsection of abstract the authors have not mentioned the results obtained from OCT studies and spectrometry.

Response 1: We have taken into account the editor's comment. (seebelow).

“(3) Results: Exposure to e-liquid in ex vivo and aerosol in vivo studies was found to result in optical clearing of lung tissue. Histological examination of lung samples showed areas of emphysematous expansion of the alveoli, thickening of the alveolar septa and the phenomenon of plasma permeation, which is less pronounced in in vivo studies than for exposure of e-liquid ex vivo. E-liquid aerosol application allows for increased resolution and improved imaging of lung tissues using OCT. Spectral studies shown significant differences between the control group and the ex vivo group in the spectral range of water absorption. It can be associated with dehydration of lung tissue owing to the hyperosmotic properties of the glycerol and propylene glycol, which are the main components of e-liquids.”

Point 2:Moreover, while you are reading the abstract, you are sure that the authors performed the study of changes of morphological and optical properties of lung tissue arising in case of vaping. However, in the introduction the authors pay attention only to tissue optical clearing without mentioning any studies concerning the morphological changes in case of vaping..

Response 2:We have taken into account the editor's comment:

“(1) Background: The use of electronic cigarettes has become widespread in recent years. The use of e-cigarettes leads to milder pathological compared to traditional cigarette smoking. Nevertheless, e-liquid vaping can cause morphological changes in the lung tissue, which effect in impaired gas exchange. This work aims on study of changes in morphological and optical properties of lung tissue under action of an e-liquid aerosol. To do this, we implemented the "passive smoking" model and created the specified concentration of aerosol of glycerol/propylene glycol mixture in the chamber with the animal.”

Point 3:As a result, they assume that the inhalation of e-liquid may be a potential diagnostic test to find out the structural changes. I recommend the authors to add more clearly explain what the exact goal of the study was and change the abstract or introduction then.

Response 3:In the study, we implemented the model of “passive smoking”, according to the literature [Kulikov V.A. Passive smoking and its consequences. Bulletin of Pharmacy. 2017; 2(76):98-102].In this model, the patient receives up to 20% of the concentration of toxic substances of an active smoker. Thus, our goal was to create the specified concentration of aerosol of glycerol/propylene glycol mixture in the chamber with the animal.We agree that it is excessive for an animal to be in such conditions; however, in this study we did not set ourselves the goal of developing a method of “preparing lung tissue for OCT examination,” but were only trying to establish whether it is, in principle, possible to achieve any effect of tissue optical clearing under the influence of the specified amount of aerosol. Only after this experiment, it became obvious that such an effect can be achieved; the development of an optimal technique for tissue preparation requires further studies that will be aimed at finding minimum aerosol concentration with a minimum time for the animal to remain in the chamber with comparable values of lung tissue opticalclearing. We changed the abstract and introduction, accordingly.

“(1) Background: The use of electronic cigarettes has become widespread in recent years. The use of e-cigarettes leads to milder pathological compared to traditional cigarette smoking. Nevertheless, e-liquid vaping can cause morphological changes in the lung tissue, which effect in impaired gas exchange.This work aims on study of changes in morphological and optical properties of lung tissue under action of an e-liquid aerosol.To do this, we implemented the "passive smoking" model and created the specified concentration of aerosol of glycerol/propylene glycol mixture in the chamber with the animal.”

“In this work, the inhalation of e-liquid was studied as a potential diagnostic test to examine pathological changes in lung tissue in vivoand ex vivo.To do this, we implemented a "passive smoking" model and created the specified concentration of aerosol of glycerol/propylene glycol mixture in the chamber with the animal.”

Point 4. The authors should more clearly explain why they have chosen such a protocol for rats who inhaled the aerosol of nicotine-free e-liquid. I agree with the reviewer, that 7.5 liters of aerosol is too much for the rat lungs.

Response 4:We have taken into account the editor's comment. The relevant fragments were added to the article.

In the study, we implemented the model of “passive smoking”, according to the literature [Kulikov V.A. Passive smoking and its consequences. Bulletin of Pharmacy. 2017; 2(76):98-102 [In Russian ]. In this model, the patient receives up to 20% of the concentration of toxic substances of an active smoker. Thus, our goal was to create the similar concentration of aerosol of glycerol/propylene glycol mixture in the chamber with the animal.

“About 3 ml of e-liquid was used in the study. There were 5 exposures of animals being placed in the inhalation chamber. On average, 0.6 ml of e-liquid was consumed per exposure. The volume of aerosol introduced into the inhalation chamber was 1500 ml, therefore the concentration of e-liquid in the aerosol was 0.004%. In the chamber, aerosol was diluted with air to 7500 ml (5 times, 20% concentration of primary aerosol in the air inhaled by the animal), therefore the concentration of e-liquid in the chamber was 0.0008%. Aerosol with such a concentration of glycerol/propylene glycol mixture was inhaled by a rat. The average minute volume of the lungs of a Wistar rat weighing about 363 g is 303 ml/min [42]. The rat spent 10 minutes in the chamber, which means that during each exposure it inhaled 3030 ml of air, and during all 5 exposuresit inhaled 15150 ml. Thus, the volume of e-liquid that passed through the rat’s lungs during the entire experiment is of 0.12 ml.

According to the model of deposition of e-liquid in a person’s lungs [41], during passive smoking through the nose, 7-10%, of the aerosol settles in the lungs. The total respiratory surface of the lungs of a white laboratory rat with weight of 363 g can be estimated as 7.5 m2 [43]. Thus, the thickness of the layer of e-liquid deposited on the surface of the rat’s lungs is 1.1-1.2 µm.”

Point 5. There also several questions about OCT study. How did the authors calculate the attenuation coefficient values? Was the whole OCT image processed or only a part of the image was used? What was the depth range for attenuation coefficient estimation? Also, are the authors sure about the absence of compression of the sample between two glass slides? If there is one, it may influence the features of morphology and, consequently, the OCT signal.

Response 5:We have taken into account the editor's comment:

“Based on the recorded OCT tomograms with entire scanning length along tissue surfaceof6 mm,the coefficient of light attenuation in the tissue was calculated by approximating the dependence of the reflected light intensity I(z) on the depth of the studied area z of the A-scan: . The attenuation coefficient was evaluated using the Mathcad software.”

“Before measurements, the samples were placed between two glass slides and fixed without compression, which did not lead to changes in tissue morphology.”

Point 6. In my opinion, the results are well described after changes that were made after reviewers’ comments. I have only several minor comments concerning the figures. The authors should add the scale bars on OCT images.

Response 6:Thank you, fixed.

Point 7. And also it would not be unnecessary to add the control histological image on the figure 3. Or it should be mentioned in the text that the control lung looks the same way as in in vivo experiments (figure 2).

Response 7:We have taken into account the editor's comment:

“Histological images of control lung samples in in vitro experiments appear similar to those in in vivo experiments(Fig.2(a)).”

Point 8. In addition, I would also recommend the authors to add they thought about the future investigation where the obtained results may be used. Because, in the current moment the discussion looks a bit incomplete.

Response 8:The additional comments have been added in the Conclusion

“The developed TOC protocols can be used in the future for in vivo lung studies using endoscopic OCT and other endoscopic optical imaging techniques. The minimal harmfulness of such protocols in in vivo studies follows from the relatively short-term exposure to aerosol from well-purified commercial e-liquids.”

Point 9. Finally, I would highly recommend the authors to carefully proofread the manuscript. Unfortunately, there are too many mistakes and typos. Especially, concerning the absence of spaces: «remainsachallenge» «remainsunder» «thevaper» «Theattenuationcoefficient».

Response 9: Thank you, fixed.

Dear Editor,

We thank you for thorough review of our manuscript. We feel that the quality of our paper has been substantially improved with the corresponding changes. We agree with your comments. We did changes. All changes are highlighted in the manuscript.

Best regards, Irina Yanina